# Bio-Mechanism of Catechin as Pheromone Signal Inhibitor: Prediction of Antibacterial Agent Action Mode by In Vitro and In Silico Study

**DOI:** 10.3390/molecules26216381

**Published:** 2021-10-22

**Authors:** Dikdik Kurnia, Zenika Febian Ramadhanty, Aprilina Mora Ardani, Achmad Zainuddin, Hendra Dian Adhita Dharsono, Mieke Hemiawati Satari

**Affiliations:** 1Department of Chemistry, Faculty of Mathematics and Natural Science, Universitas Padjadjaran, Sumedang 45363, Indonesia; zenikafr@gmail.com (Z.F.R.); aprilinamoraardani@gmail.com (A.M.A.); a.zainuddin@unpad.ac.id (A.Z.); 2Department of Conservative Dentistry, Faculty of Dentistry, Universitas Padjadjaran, Bandung 40132, Indonesia; adhita.dharsono@fkg.unpad.ac.id; 3Department of Oral Biology, Faculty of Dentistry, Universitas Padjadjaran, Bandung 40132, Indonesia; mieke.satari@unpad.ac.id

**Keywords:** catechin, QS, GBAP, gelatinase, MurA, serine protease

## Abstract

The utilization of medicinal plants has long been explored for the discovery of antibacterial agents and the most effective mechanisms or new targets that can prevent and control the spread of antibiotic resistance. One kind of bacterial cell wall inhibition is the inactivation of the MurA enzyme that contributes to the formation of peptidoglycan. Another approach is to interfere with the cell–cell communication of bacteria called the Quorum sensing (QS) system. The blocking of auto-inducer such as gelatinase biosynthesis-activating pheromone (GBAP) can also suppress the virulence factors of gelatinase and serine protease. This research, in particular, aims to analyze lead compounds as antibacterial and anti-QS agents from Gambir (*Uncaria gambir* Roxburgh) through protein inhibition by in silico study. Antibacterial agents were isolated by bioactivity-guided isolation using a combination of chromatographic methods, and their chemical structures were determined by spectroscopic analysis methods. The in vitro antibacterial activity was evaluated by disc diffusion methods to determine inhibitory values. Meanwhile, in the in silico analysis, the compound of *Uncaria gambir* was used as ligand and compared with fosfomycin, ambuic acid, quercetin, and taxifolin as the standard ligand. These ligands were attached to MurA, GBAP, gelatinase, and serine proteases using Autodock Vina in PyRx 0.8 followed by PYMOL for combining the ligand conformation and proteins. plus programs to explore the complex, and visualized by Discovery Studio 2020 Client program. The antibacterial agent was identified as catechin that showed inhibitory activity against *Enterococcus faecalis* ATCC 29212 with inhibition zones of 11.70 mm at 10%, together with MIC and MBC values of 0.63 and 1.25 μg/mL, respectively. In the in silico study, the molecular interaction of catechin with MurA, GBAP, and gelatinase proteins showed good binding energy compared with two positive controls, namely fosfomycin and ambuic acid. It is better to use catechin–MurA (−8.5 Kcal/mol) and catechin–gelatinase (−7.8 Kcal/mol), as they have binding energies which are not marginally different from quercetin and taxifolin. On the other hand, the binding energy of serine protease is lower than quercetin, taxifolin, and ambuic acid. Based on the data, catechin has potency as an antibacterial through the inhibition of GBAP proteins, gelatinase, and serine protease that play a role in the QS system. This is the first discovery of the potential of catechin as an alternative antibacterial agent with an effective mechanism to prevent and control oral disease affected by antibiotic resistance.

## 1. Introduction

Oral health is directly related to our general health as well as quality of life [1]. According to the World Health Organization (WHO), 60–90% of children and almost 100% of adults have dental caries and 15–20% of middle-aged people become infected with periodontal diseases [2]. Dental caries is the process of demineralization and remineralization of enamel, dentin in the fissures, and the smooth surfaces of the tooth caused by oral bacteria [3]. Periodontal disease is a bacterial infection that damages the supporting structure of the tooth. *Streptococcus mutans, Streptococcus sanguinis,* and *Enterococcus faecalis* are the pathogenic bacteria that cause oral diseases [4,5]. Five target mechanisms of antibacterial pathways have been clinically validated, including inhibition of cell wall synthesis, inhibition of protein synthesis, inhibition of DNA or RNA synthesis, inhibition of folate synthesis, and disruption of cell membranes [6]. The presence of a large number of drug-resistant bacteria has led to the discovery of antibacterial therapy with new mechanisms or targets. One promising mechanism is the inhibition of bacterial cell wall synthesis [7].

The cell wall of bacteria is one of their defenses against the external environment, consisting of polysaccharides, polypeptides, and peptidoglycan. The role of peptidoglycan is to stabilize the cell walls as large cellular macromolecules or major structural components. Subsequently, peptidoglycan biosynthesis is a multi-step process involving: (1) formation of *N*-acetylglucosamine (NAG) and *N*-acetylmuramic acid (NAM) disaccharide pentapeptide peptidoglycan precursors; (2) transport across the cell envelope, and (3) assembly of the precursors into the growing peptidoglycan layer. Mur enzymes, particularly MurA enzyme, are one of the enzymes required for the synthesis of the disaccharide pentapeptide. UDP-*N*-acetylglucosamine-1-carboxyvinyltransferase (MurA) catalyzes the first step of peptidoglycan precursor biosynthesis [8]. Therefore, the inhibition of MurA enzyme is considered a step to prevent bacterial survival, as it plays an important role in both Gram-positive and Gram-negative bacteria [9].

Another approach to inhibit bacterial growth is to disrupt the Quorum sensing (QS) system. This method has been widely considered in evaluations for the development of new therapies to deal with antimicrobial resistance. The QS systems involve bacteria communication that can control gene expression by signaling molecules or auto-inducers [10]. In sensing the density of the cell quorum, the latter will cause the hardening of molecules in the environment and help activate the appropriate receptors on the cell surface or cytoplasm. Consequently, the transcription of the QS gene will be active [11]. One of the genes allows for suppressing virulence and pathogenicity factors which can be encoded by the fsr locus of *Enterococcus faecalis* bacteria [12,13,14]. Among the important fsrABCD genes, the fsrD gene is responsible for gelatinase biosynthesis-activating pheromone (GBAP) expresion. The active signal will be transmitted by other fsr genes, and the fsr control system will control the expression of protease gelatinase (GelE) and serine protease (SprE) [15,16,17]. By blocking one of the QS stages such as GBAP inhibition, the cell–cell communication in bacteria will be ruined and the bacteria will die automatically [18]. The other approach is to interfere with communication between bacterial cells and a QS system, facilitated by using computational and docking methods. Virtual filtering can help find the most appropriate set of compounds that show bioactivity from natural resources in an effective manner [19,20].

Currently, the research and development of new drugs from natural products is becoming an interesting topic among many researchers across the world, as natural products are one of the important sources of bioactive compounds. To further continue our search for new prospective antibacterial agents against pathogenic oral bacteria, we attempted to identify and determine the antibacterial components of a medicinal plant, Gambir *(Uncaria gambir* Roxburgh). This plant is a member of *Rubiaceae* family [21] and the main commodity produced in West Sumatra, Indonesia [22]. The word Roxburgh is taken from the name of a botanist who came from Scotland, William Roxburgh. *Uncaria gambir* Roxb. is commonly used as traditional medicine for diarrhea and as a chewing refreshment [23]. It contains phenolics [24], alkaloids, flavonoids [25], and tannins [26] that exhibit antibacterial and antioxidant properties [27]. In this study, we focus on finding the antibacterial constituents of *Uncaria gambir* Roxb. against pathogenic oral bacterium *E. faecalis* through the most effective mechanism by predicting the molecular interaction in inhibiting the GBAP protein, gelatinase, and serine proteases that play a role in the QS system.

## 2. Results

### 2.1. Chemical Composition of Extract Uncaria gambir Roxb.

The phytochemical screening of the fruit of *U. gambir* Roxb. shows the chemical content qualitatively in each extract, including methanol and ethyl acetate extracts, containing secondary metabolites of flavonoids, triterpenoids, tannins, saponins, and alkaloids. The water extract contains flavonoids, tannins, saponins, and alkaloids. Meanwhile, *n*-hexane extract contains triterpenoids and steroids.

### 2.2. Structure Determination of Compound

Compound **1** was isolated in purple powder which dissolves in methanol. Based on the infrared measurement, the spectrum showed absorption peaks at 3305 cm^−1^ corresponding to the hydroxyl group. Other peaks were observed at 2935, 1627, 1520, and 1113 which were identified as signals for the presence of C-H *sp*^3^, C-C, C=C, and C-O groups, respectively. The UV spectra of **1** showed an absorption maximum at 254 and 279 nm presented by a band I flavonoid compound [28].

Based on ^13^C-NMR and DEPT 135° spectra, compound **1** contained fifteen carbons corresponding to one of *sp*^3^ carbon at δc 28.5 ppm and two oxygenated carbons at δc 68.8, and 82.8 ppm, respectively, along with twelve *sp*^2^ aromatic carbons at δc 95.5–157.8 ppm. These signals were identified as one methylene, seven methines, and seven quaternary carbons. Based on the carbon assignment, compound **1** was suggested to have a carbon skeleton of the flavan group [29].

Further analysis of the ^1^H-NMR spectra showed signals addressed to two protons at δ_H_ 2.51 (1H; dd; *J* = 8.1; 16.1) and 2.85 (1 H; dd; *J* =8.1; 16.1) bound to *sp*^3^ carbon, two protons at δ_H_ 3.98 (1H; m) and 4.57 (1H; d; *J* = 7.4) bound to oxygenated *sp*^3^ carbon, and one hydroxyl group at δ_H_ 3.35 (s; br) bound to *sp*^3^ carbon. These signals suggested that the compound is a part of the flavan carbon group. Other signals were identified for five protons at δ_H_ 6–8 ppm bound to methine carbon (-CH=) of a flavan group, two aromatic protons at δ_H_ 5.86 (1H; d; *J =* 2.3) ppm and δ_H_ 5.93 (1H; d; *J =* 2.3) ppm as *meta*-position in Ring A. Then, we identified protons at δ_H_ 6.73 (1H; d; *J =* 8.15 and 2.05) corresponding to *ortho* position and an aromatic proton at δ_H_ 6.77 (1H; d; *J =* 8.15) as *meta*-position in Ring B of the flavan group. The peak of the mass spectroscopic spectrum was obtained at *m*/*z* 289.13 [ES^−1^] with the molecular formula C_15_O_6_H_14_. The result of the double-bound equivalent (DBE) calculation was nine including two benzenes and a cyclohexane group. Based on the analysis of spectroscopic data compared to published papers, compound **1** was identified as catechin [30].

Spectral data of catechin (**1**): UV (MeOH) λ_max_ 205, 254 and 279 nm; IR (KBr): 3305, 2935, 1627, 1520 and 1113 cm^−1^. The MS: (*m*/*z*) 289.13 [ES^−1^] for (C_15_O_6_H_14_). ^1^H-NMR (CD_3_OD): δ_H_ 4.57 (1H, d, 7.4, H-1); 3.98 (1H, *m*, H-2); 2.51 (1H, dd, 8.1, 16.1, H-3_a_) & 2.85 (1H, dd, 5.4, 16.1)H-3_b_); 5.93 (1H, d, 2.3, H-6); 5.86 (1H, d, 2.3, H-8); 6.84 (1H, d, 2.0, H-11); 6.77 (1H, d, 8.15, H-14); 6.73 (1H, dd, 8.15, 2.05, H-15). ^13^C-NMR (CD_3_OD): δ_C_ 82.8 (C-1), 68.8 (C-2), 28.5 (C-3), 100.8 (C-4), 157.8 (C-5), 96.3 (C-6), 157.5 (C-7), 95.5 (C-8), 156.9 (C-9), 132.2 (C-10), 115.2 (C-11), 146.2 (C-12), 146.3 (C-13), 116.1 (C-14), and 120.1 (C-15). DEPT 135°: methylene (C-3), methin (C-1, C-2, C-6, C-8, C-11, C-14, and C-15), quaternary carbon (C-4, C-5, C-7, C-9, C-10, C12, and C-13).

### 2.3. Antibacterial Activity Assay of Extracts against E. faecalis ATCC 29212

To evaluate the potential antibacterial activity, the extracts were assayed to determine the inhibition zone values against *E. faecalis* using the Kirby–Bauer method. The data of antibacterial extracts are reported in Table 1.

### 2.4. Determining the Antibacterial Activity (MIC and MBC) of Compound

To evaluate the antibacterial activity of compound **1**, the inhibition zone values were measured by the Kirby–Bauer method. As shown in Table 2, catechin (**1**) inhibited the growth of *E. faecalis* at 10% which is equal to the inhibition zones of chlorhexidine at 2% as a positive control. According to the criteria of inhibition zone values, catechin (**1**) at 10% with inhibition zones of 11.7 mm was classified as a strong antibacterial agent (i.z ≥6 mm: strong, 3–6 mm: moderate, ≤3 mm: weak) [31].

In further antibacterial activity evaluation, the minimum inhibition concentration (MIC) and minimum bactericidal concentration (MBC) were determined. As shown in Table 3, the active antibacterial compound catechin (**1**) showed MIC and MBC values of 0.625 and 1.25 μg/mL, respectively. Based on the data in published papers, the MIC and MBC values of **1** were categorized as good antibacterial activity [32]. On the other hand, fosfomycin as a control showed an MIC value of 62.5 μg/mL and no MBC value [33]. This means that fosfomycin simply inhibits bacterial growth but does not kill the bacteria. Based on the MIC and MBC values, it was suggested that the antibacterial activity of catechin (**1**) is better than fosfomycin and chlorhexidine as a positive control and we assumed that catechin (**1**) has the potency as a new antibacterial agent against oral pathogenic bacteria.

### 2.5. Antibacterial Activity Prediction from Uncaria gambir Roxb. through Molecular Docking

Based on the analysis data of in vitro antibacterial activity of catechin against *E. faecalis*, the molecular mechanism prediction was designed and evaluated by in silico study. In this docking process, we used several proteins involved in bacterial cell wall inhibition, such as MurA enzymes and proteins involved in the QS, including gelatinase, and serine protease. The result of the docking analysis is shown in Table 4 and Table 5.

The data analysis presented in Table 4 showed that catechin has the third strongest position for binding affinity in each protein. Binding affinities of catechin in MurA, GBAP, gelatinase, and serine protease are −8.5, −5.2, −7.8, and −7.0 Kcal.mol^−1^, respectively. Generally, the strongest binding affinity is shown by quercetin against three proteins in the QS system (−5.2 Kcal.mol^−1^ for GBAP, −8.3 Kcal.mol^−1^ for gelatinase, and −6.9 Kcal.mol^− 1^ for serine protease). However, in the MurA enzyme, taxifolin is the best ligand with a value of −9.1 Kcal.mol^−1^.

Table 5 indicates that amino acid residues bound to targeted proteins were different. In MurA enzyme and gelatinase, all ligands were attached to the same pocket (see Figure 1 and Figure 2) but have different residues. There were different residues attached by flavonoid and non-flavonoid ligands. Catechin, quercetin, and taxifolin bonded to Gly164A in MurA while fosfomycin and ambuic acid did not attach to it. On the other hand, fosfomycin and ambuic acid bind Arg384A in gelatinase whereas flavonoid-shape ligands did not attach to it.

Unlike in MurA and gelatinase, in GBAP (see Figure 3) almost all ligands except catechin and taxifolin were attached in a different position. The catechin and taxifolin bind Arg15A and Ser33A. Meanwhile, there were three ligation positions in serine protease (see Figure 4). The first was the position attachment of fosfomycin and ambuic acid, the second was of catechin and quercetin, and the last was of taxifolin, respectively.

## 3. Discussion

Bacterial infection by pathogenic oral bacteria such as *E. faecalis* causes oral infections including caries and periodontal disease. The use of some antibiotics and antibacterial agents to prevent and cure such infections over a long period, however, has resulted in side effects of resistance to these agents. To solve this, new antibacterial agents are important research targets. Since the sources and structures of active compounds are important to investigate, the receptor or protein molecular target of the new drug should also be considered a critical aspect of active molecules that can effectively act as a new antibacterial agent. Moreover, it is important to achieve an appropriate molecular interaction between active compounds and the receptor target at the final condition.

To find a new potential antibacterial agent against oral pathogens from natural sources, the herbal plant Gambir (*Uncaria gambir* Roxb.) was selected and bioactivity-guided isolation was carried out (more information in Appendix A). The first step was the selection of the most active among the methanolic extracts of n-hexane, ethyl acetate, and air. Furthermore, the compound was isolated from the most active extract. Table 1 shows that ethyl acetate, methanol, and water fractions were able to inhibit the growth of *E. faecalis* with 10% concentration and inhibition zone values of 8.30, 7.65, and 6.75 mm. On the other hand, chlorhexidine 2% as a positive control has an inhibition zone of 16.40 mm. Further separation and purification of ethyl acetate to isolate its active antibacterial constituents validated that catechin (**1**) is an active compound with good MIB and MBC values as compared to chlorhexidine and fosfomycin as a positive control (Table 3). In a published report, researchers claimed catechin as an antioxidant and antibacterial agent [34], however, the molecular mechanism of bacterial inhibitions by interfering with bacterial communication signal or their QS system has not been reported yet. To examine its mechanism, the molecular modeling approach for in silico experiment of catechin was performed and supported by preliminary in vitro antibacterial assay data along with some positive controls of antibacterial compounds as references.

The results of molecular docking helped see the mechanism of interaction between catechin and UDP-*N*-acetylglucosamine enolpyruvyl transferase (MurA). Additionally, the higher binding affinity between proteins and ligands helped the drugs show a higher efficacy [35]. The lower the binding energy, the stronger and more stable is the bond [36]. In this virtual screening, fosfomycin, the MurA enzyme inhibitor, was used as a positive control because it can inhibit the biosynthesis of bacterial cell walls, facilitating bactericidal activity against anaerobic pathogens as well as many Gram-positive and Gram-negative bacteria [37].

The docking results in Table 4 and Table 5 showed that the hydrogen bond of catechin and MurA is preferred, as it has a binding energy of −8.5 Kcal/mol. Additionally, the catechin–gelatinase bond and catechin–protease have a fairly good binding affinity, but the catechin-GBAP bond is the weakest with the binding affinity of −5.0 Kcal/mol. Fosfomycin as one of the positive controls has weaker binding energy than catechin. Consequently, catechin is considered a potential MurA inhibitor owing to its binding affinity of −8.5 Kcal/mol which is better than two positive controls, such as fosfomycin and ambuic acid, and almost equal to that of quercetin, −8.5 Kcal/mol. Meanwhile, the catechin binding to GBAP and gelatinase is strong enough (greater than those of the two positive controls, fosfomycin and ambuic acid). Therefore, they are suggested as GBAP and gelatinase inhibitors. On the other hand, the bonds of catechin–serine proteases are better than fosfomycin (−7.0 Kcal/mol) and the other three positive controls.

In Figure 1, the MurA enzyme has several bonds in the same residue against some proteins including catechin, fosfomycin, and taxifolin. These ligands bind to the Asp 305 residue in the MurA enzyme. The involved residues include Asp305, Arg93, and Val327. Therefore, the inhibition of these key residues will inhibit the normal functioning of the enzyme [38]. Besides, catechin also binds to the residues Ser162A and Gly164A as well as the positive control of quercetin and taxifolin. In the docking result of the MurA enzyme, fosfomycin binds to Asp305 and acts as a MurA enzyme inhibitor. However, the binding affinity remains the lowest among other ligands. The hydrogen bonds that are formed between the ligands and proteins will contribute to ligand-protein stability. The hydrogen bond will show a green bond in the picture, while a bond in purple indicates a hydrophobic bond. The strength of the hydrogen bond can also be built by the distance of the ligand and protein. The smaller the ligand-protein distance, the stronger the binding energy [39].

In the gelatinase–ligand complex (Figure 2), all ligands are almost bound to the same residue. His332A and Asn298A residues are particularly bound to catechin, quercetin, and taxifolin. The Glu329A residue is bound to catechin, fosfomycin, and quercetin ligands. Tyr343A residue, on the other hand, is only bound to catechin and quercetin. Fosfomycin and ambuic acid have weak binding affinity, as it is not bound to the residual His332A and Tyr343A that play an important role in gelatinase protein.

In the GBAP–ligand complexes (Figure 3), the bound residues are not too competitive; catechin and taxifolin bind to the same residue (Arg15 and Ser33), while fosfomycin and taxifolin bind to Gln31A. Ambuic acid only binds to the residue of Ile36 with a bond affinity of −4.5 Kcal/mol. Therefore, it is placed in the second position—the weakest after fosfomycin.

In serine proteases (Figure 4), catechin has the same bonds as quercetin (Asn145A, Asn262A, Asn268A) with the highest binding energy of −6.9 Kcal/mol. It is not marginally different from taxifolin and ambuic acid, while fosfomycin has the lowest value of −3.9 Kcal/mol.

Based on the data obtained, catechins are the potential anti-QS agents, as they have a competitive binding affinity compared to positive controls used for GBAP protein, gelatinase, and serine protease. With inhibition of catechin against gelatinase and serine protease, the regulation of the fsr system will be disrupted. Consequently, the QS system will be inhibited by signaling molecules (GBAP) so that the formation of GBAP will be disrupted indirectly. In addition, catechin suppresses bacteria phatogenicity by reducing virulence factor gelatinase and serine protease.

Based on all experimental data, catechin acts as a non-competitive inhibitor with non-flavonoid ligands against MurA, GBAP, gelatinase, and serine protease. However, it competes with flavonoid ligands, except for serine protease.

## 4. Materials and Methods

### 4.1. Plant Materials and Chemicals

Gambir (*Uncaria gambir* Roxb.) is an endemic plant from Sumatera Island, Indonesia. The plant material was collected in June 2017 from a local farmer in Padang, West Sumatra, Indonesia. The plant specimen was identified at the laboratory of Taxonomy, Department of Biology, Faculty of Mathematics and Natural Science, Universitas Padjadjaran, Sumedang, Indonesia. For the extraction process, separation and purification were performed using distilled organic solvents of methanol, *n*-hexane, and ethyl acetate, while the analytical-grade organic solvents from Merck Co. Ltd. and Sigma Aldrich Co. Ltd. (St. Louis, MO, USA) were used for spectroscopic analysis. Silica G 60 (Merck, Darmstadt, Germany) and ODS RP-18 were used for column chromatography (CC), while silica G 60 F_254_ and ODS RP-18 F_254S_ (Merck, Darmstadt, Germany) plates were used for thin-layer chromatography (TLC). The spot compounds on TLC were visualized under UV light at 254 and 356 nm by spraying with 10% H_2_SO_4_ in EtOH followed by heating.

### 4.2. Instruments

The structure of active compounds was determined by spectroscopic methods of ultra-violet (UV) by 8452A Diode Array (Hewlett Packard Palo Alto, CA, USA), infra-red (IR) with FTIR Shimadzu 8400 (SpectrLab Scientific Inc., Markham, Canada), NMR (^1^H-NMR, ^13^C-NMR, and DEPT 135°) with JEOL type ECA 500 MHz (JEOL ltd., Tokyo, Japan), and mass spectrometry (MS) with Water Acquit UPLC type triquadrupole (Agilent, CA, USA). The TLC plates were visualized by UV detector lamps with wavelengths of λ_max_ at 254 and 365 nm. Antibacterial activity assay used 96-well microplates (NEST Biotechnology, Wuxi, China), micropipettes (Winlab, Grogol, Jakarta), microtubes (Chemikalie, Pandan Loop, Singapore), incubators (Memmert, Schwabach, Germany), paper disks (Grainger approved, Origin, USA), and Biochrom microplate readers (Biochrom ltd., Cambridge, UK).

### 4.3. Materials for Antibacterial Activity Assay: In Vitro Study

The antimicrobial assay used *Enterococcus faecalis* ATCC 29212, Muller Hinton broth, and Muller Hinton agar as a medium and chlorhexidine as the positive controls.

### 4.4. Materials for Molecular Docking: In Silico Study

The 3D structure of MurA used in this study was obtained from Protein Data Bank (PDB) ID: 1UAE [40]. A protein (enzyme) UDP-*N*-acetylglucosamine enolpyruvyl transferase (MurA) from UniProt Knowledgebase (http://www.uniprot.org/): UniProtKB-P0A749 was used as a receptor. The 3D structure of MurA was obtained using the RSCB program (https://www.rcsb.org) with 1UAE format PDB, and the 3D structure model of GBAP G8ADP0, gelatinase Q833V7, and serine proteinase A1YGV8 were retrieved from UniProt Knowledgebase (http://www.uniprot.org/). The MurA enzyme produced from the Swiss-Model using template 1UAE has a sequence identity of 100% with GMEAN values of 0.95 and QMEAN −0.52 which indicate good model quality. Gelatinase enzyme using template 5A3Y has a sequence identity of 28.64% with a GMEAN value of 0.35 and QMEAN −3.67. Lastly, the serine protease enzyme using template 6U1B has a sequence identity of 26.19% with a GMEAN value of 0.46 and QMEAN −4.05. A compound of *Uncaria gambir* Roxb. called catechin is the tested ligand. It was retrieved from Pubchem with CID compound–CID 9064 [41]. Ligands as positive controls are fosfomycin (CID 446987) [42], ambuic acid (CID 11152290), quercetin (CID 5280343), and taxifolin (CID 439533) [43]. All these data were taken from the PubChem compound database (https://www.ncbi.nlm.nih.gov/pccompound) and presented in Figure 5.

### 4.5. Experimental

#### 4.5.1. Preparation of Extracts of Gambir (*Uncaria gambir* Roxb.) Fruit

*Uncaria gambir* Roxb. fruit (1.25 kg) was cut into small pieces and macerated using methanol solvent for 3 × 24 h [44]. The result of maceration was filtrated and the filtrate was concentrated with a rotary evaporator at 40 °C. Then, the sample was partitioned with water-*n*-hexane and water-ethyl acetate and subsequently partitioned between *n*-hexane-water and ethyl acetate-water. To get extracts of methanol (269.6 g), *n*-hexane (3.22 g), and ethyl acetate (161.16 g), and water (10.32 g), the sample was concentrated with a rotatory evaporator and made in a series concentration of 10% for phytochemical screening and antibacterial preliminary bioactivity evaluations against *E. faecalis* ATCC 29212.

#### 4.5.2. Phytochemical Screening

Secondary metabolite content in *Uncaria gambir* was analyzed by qualitative analysis of phytochemical screening by reacting the sample solution with several specific reagents. The existence of phenol was analyzed by adding FeCl_3_ 5% to sample. HCl and powder of Mg was added to sample solution to identify flavonoids, steroids, and triterpenoids, which were analyzed by reacting sample solution with Liebermann Burchard reagent. Saponin was analyzed by adding hot water to the sample then adding HCl 2N to the solution. Tannin was analyzed by reacting sample solution with FeCl_3_ 1%. Dragendroff reagent was added to sample solution to identify alkaloids.

#### 4.5.3. Isolation of Compound from Ethyl Acetate Extract of *Uncaria gambir* Roxb. Fruit

The ethyl acetate extract (161.16 g) was separated by column chromatography method with silica gel G60 stationary phase and 5% mobile phase of *n*-hexane and ethyl acetate. From this separation, a fraction with simple compound composition was obtained, guided by thin-layer chromatography (TLC). Then, it was separated again using the same method with ODS RP-18 stationary phase and eluted with methanol and water in a gradient of 5% and 10% mobile phase. A simple compound was obtained guided by TLC. The third separation with silica gel G60 stationary phase, eluted by *n*-hexane and ethyl acetic in a gradient of 1% and 5% mobile phase, resulted in pure compound 1 (24 mg). The purity of 1 was analyzed by performing 1D- and 2D- TLC with the combination of normal and reverse phase TLC plates of Silica G 60 F_254_ and ODS RP-18 F_254_S, respectively.

#### 4.5.4. Structure Determination of Active Compound **1**

The structure of active compounds was determined using comprehensive analysis data of spectroscopic methods including ultraviolet (UV) spectrum, infrared (IR) spectrum, 1D and 2D-NMR spectra (^1^H-NMR, ^13^C-NMR, DEPT 135°, HMQC, ^1^H-^1^H COSY, HMBC), and mass spectrometry (MS).

#### 4.5.5. Evaluation of the Antibacterial Activity of the Extract and Active Compound of *Uncaria gambir* Roxb.

##### Microorganism Assay

The bacteria *E. faecalis* ATCC 29212 was used for the antibacterial test with Muller Hinton broth and Muller Hinton agar as mediums and chlorhexidine as a positive control. Additionally, tube anaerobic (purchased from Merck Co. Ltd. and Sigma Aldrich) was used in the antibacterial assay.

##### Antibacterial Activity

The antibacterial effect of the extract and compound **1** against *E. faecalis* was determined using Kirby–Bauer disk diffusion. This also allows determining the sensitivity of pathogenic oral bacteria [45]. Extract and compound were diluted with methanol–water (1:1) and chlorohexidine as positive control was diluted with water. Extract was adjusted in 10% while compound was adjusted to various concentrations of 10%, 5%, 2%, and 1%. Next, 20 µL of the sample was placed in the paper disc (6 mm) and was placed on the surface of the agar. The test was performed in duplicate.

The MIC and MBC activity of compound **1**
*Uncaria gambir* Roxb. against *E. faecalis* ATCC 29212 was determined by the microdilution method on a 96-well microplate [46]. The bacterial cells were pre-cultured in Muller Hinton broth at 37 °C under aerobic conditions. They were then incubated in the presence of compounds with the concentrations obtained by serial two-fold dilution at 37 °C without shaking the same broth for 24 h in microplate wells. The optical density of the solution in the microplate was measured using a microplate of 620 nm. The MICs were estimated as the lowest concentrations where the bacterial cells were observed by OD value. Then, each solution in the well was spread on the surface of the agar and incubated for 24 h at 37 °C. The minimum concentration of the sample, i.e., no bacteria growth under colony counter, was determined as the MBC value.

#### 4.5.6. In Silico Characterization of the *Uncaria gambir* Roxb. Compounds

The characteristics of catechin were confirmed using an online program. Its chemical structure was obtained from PubChem (https://pubchem.ncbi.nlm.nih.gov/compound/292101) by downloading Canonical SMILES format (C1C(C(OC2=CC(=CC(=C21)O)O)C3=CC(=C(C=C3)O)O)O). Canonical SMILES is used to convert chemical structures into 3D using the OPEN BABEL 2.4.2 program in PDB format [47]. Furthermore, the 3D structure of UDP-N-acetylglucosamine enolpyruvyl transferase (MurA) was obtained using the RCSB PDB (https://www.rcsb.org/structure/1UAE) in PDB format. The 3D structure model of GBAP, gelatinase, and serine protease was built using the SWISS-MODEL server (https://swissmodel.expasy.org/) in PDB format [48].

#### 4.5.7. Molecular Docking between MurA/GBAP/Gelatinase/Serine Protease and *Uncaria gambir* Roxb. Compound

Virtual screening and docking of ligand-protein used Autodock Vina in the open-source PyRx 0.8 software [49]. Catechin as a ligand was used to bind four proteins (MurA, GBAP, gelE, and sprE) as a protein target, ligand-free for blind docking. The docking process started by selecting the macromolecule and four ligands. Moreover, the step-by-step process was followed as manual instruction until the binding energy of the macromolecule–ligand was calculated. The selected conformation was the conformation with the lowest bond energy of value less than 1.0Å in the mean square root deviation (RMSD).

#### 4.5.8. Complex MurA/GBAP/Gelatinase/Serine Protease—Catechin Visualization and Analysis

The conformations were combined with protein targets using PYMOL. The complex of ligand-residue was analyzed by online program Proteins Plus (https://proteins.plus/) [50] and visualized by Discovery Studio 2020 Client program. The PYMOL program shows the docking position and ligand-residue interactions in the form of 3D molecules. Amino acids that bind to the ligand of each ligand-residue complex were compared with the 3D structure of a protein that binds to the ligands on catalytic sites of each protein (binding site of Fosfomycin for MurA, ambuic acid for GBAP). This step was used to determine the similarity of the catechin ligation position with fosfomycin.

## 5. Conclusions

Catechin from *Uncaria gambir* Roxb. fruit has good activity to inhibit *E. faecalis* and consequently presented as an anti-QS and antibacterial agent. The binding affinity of catechin is stronger than the original ligand of each protein (fosfomycin for MurA and ambuic acid for GBAP). Moreover, the attachment positions of catechin are different from both of these proteins. Therefore, catechin can be an alternative antibacterial agent and increase the inhibition if used with positive controls. This study can be used as basic data to support further analysis and the development of new antibacterial agents through further specific and selective in vitro, in vivo, and clinical studies.

## Figures and Tables

**Figure 1 molecules-26-06381-f001:**
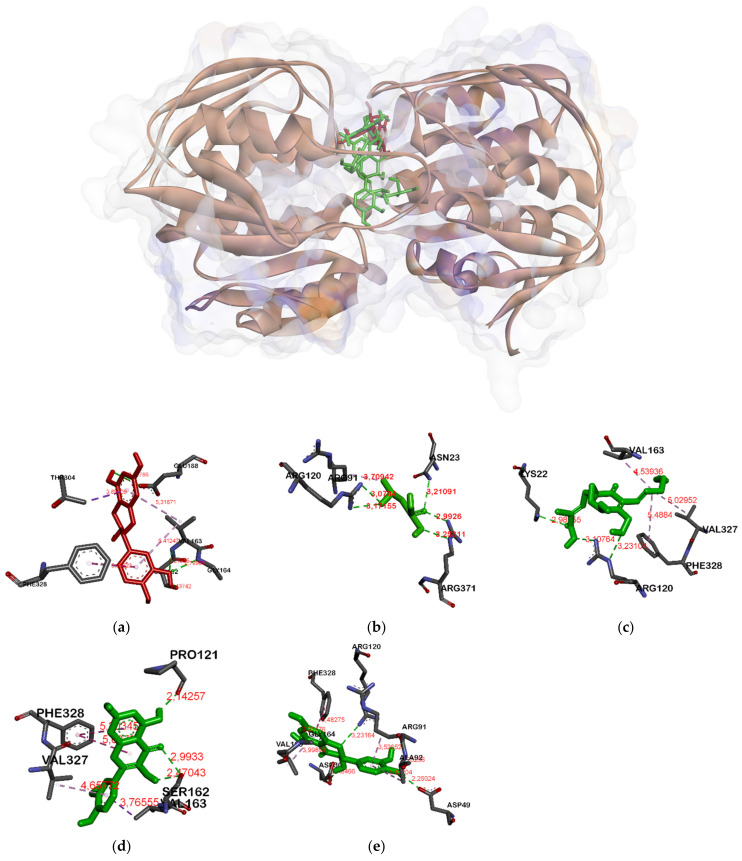
Binding site on MurA for catechin (**a**), fosfomycin (**b**), ambuic acid (**c**), quercetin (**d**) and taxifolin (**e**).

**Figure 2 molecules-26-06381-f002:**
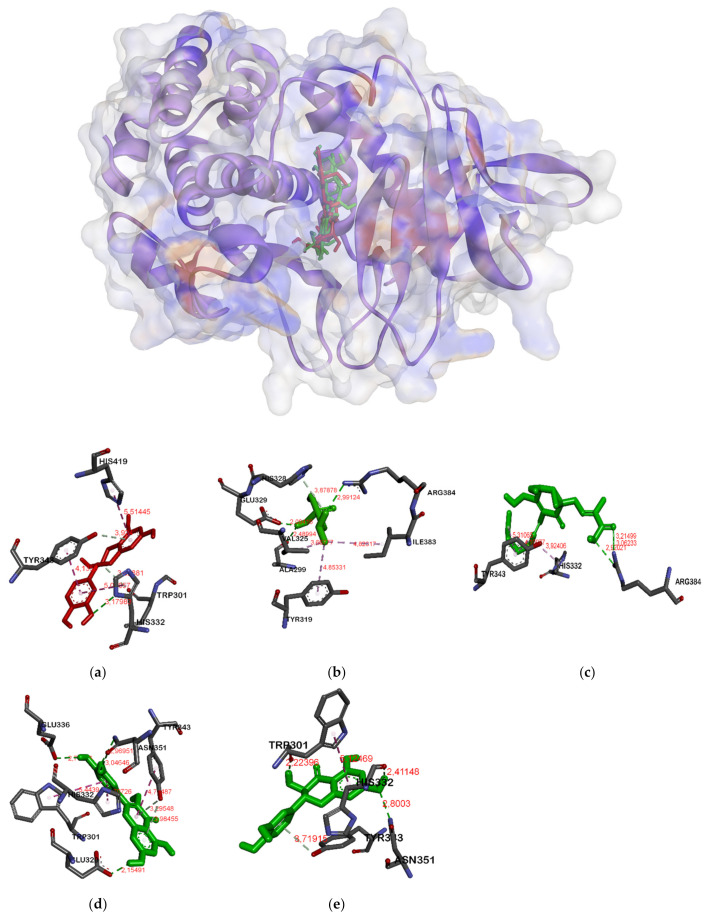
Binding site on gelatinase for catechin (**a**), fosfomycin (**b**), ambuic acid (**c**), quercetin (**d**) and taxifolin (**e**).

**Figure 3 molecules-26-06381-f003:**
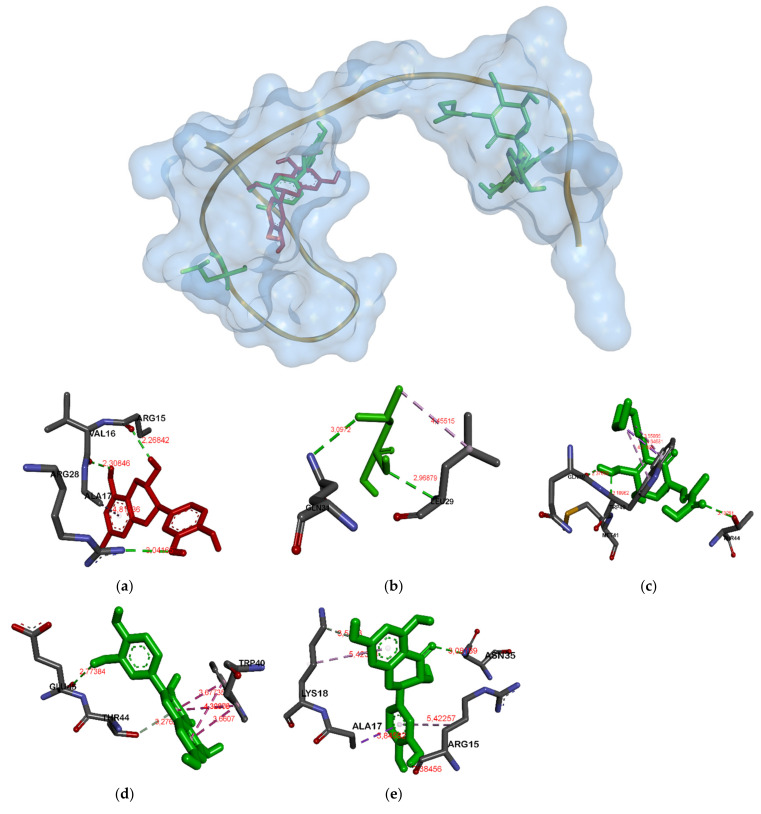
Binding site on GBAP for catechin (**a**), fosfomycin (**b**), ambuic acid (**c**), quercetin (**d**) and taxifolin (**e**).

**Figure 4 molecules-26-06381-f004:**
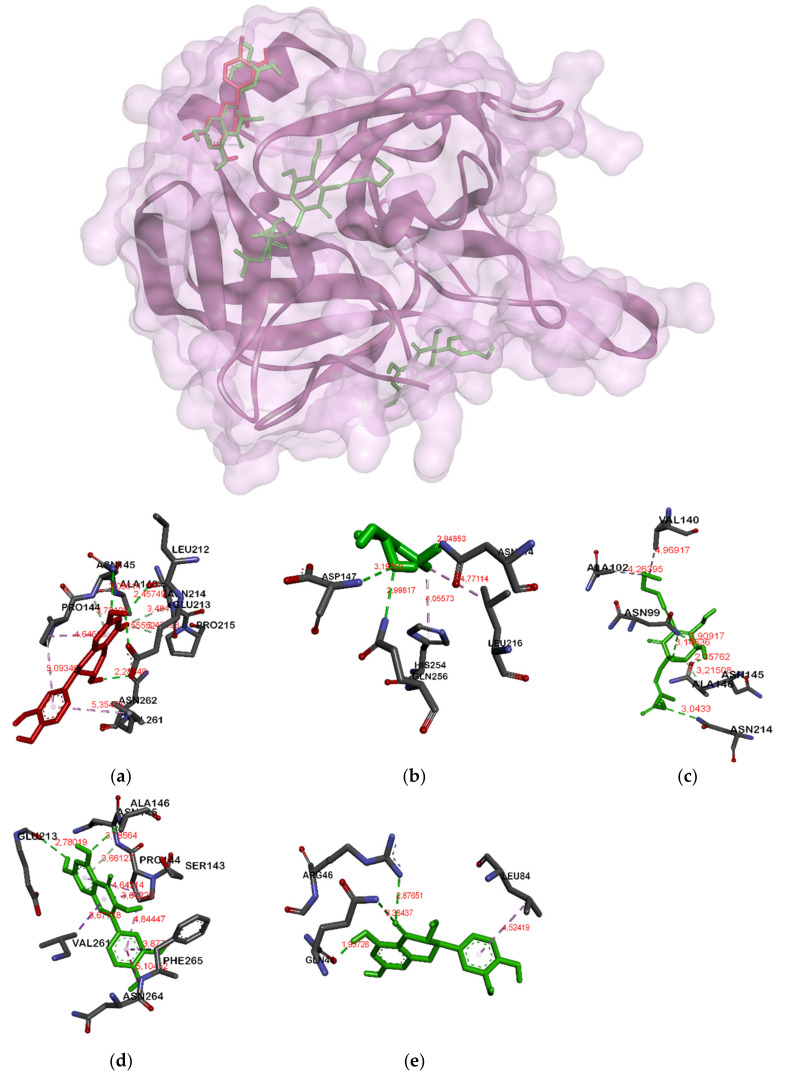
Binding site on serine protease for catechin (**a**), fosfomycin (**b**), ambuic acid (**c**), quercetin (**d**) and taxifolin (**e**).

**Figure 5 molecules-26-06381-f005:**
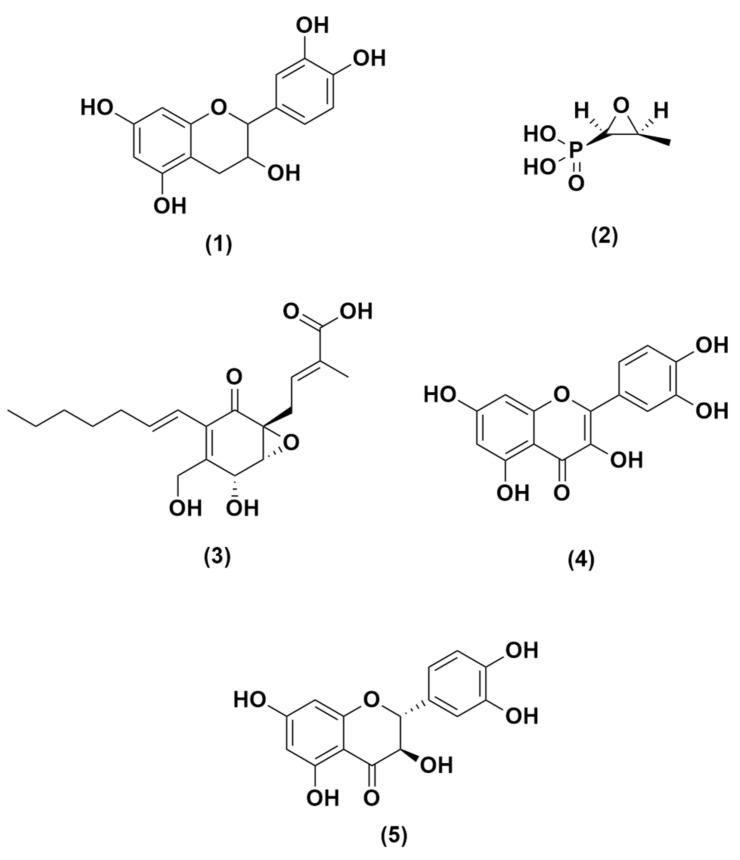
Structure of compounds of catechin (**1**) from Gambir (*Uncaria gambir* Roxb.); positive control: fosfomycin (**2**), ambuic acid (**3**), quercetin (**4**) and taxifolin (**5**).

**Table 1 molecules-26-06381-t001:** The inhibition zones of *Uncaria gambir* Roxb. Extracts against *E. faecalis* ATCC 29212.

No	Extracts	Concentration (*%*)	Inhibition Zone (mm)
1	*n*-Hexane	10	0.00
2	Ethyl acetate		8.30
3	Methanol		7.65
4	Water		6.75
5	Chlorhexidine	2	16.40

Chlorhexidine (CHx) was used as a positive control.

**Table 2 molecules-26-06381-t002:** The inhibition zones of compound **1** against *E. faecalis* ATCC 29212.

No.	Compounds	Concentrations (%)	Inhibition Zones (mm)
1	Catechin (**1**)	10	11.7
2	Chlorhexidine	2	26.5

Chlorhexidine (CHx) was used as a positive control.

**Table 3 molecules-26-06381-t003:** MIC and MBC data of Catechin against *E. faecalis* ATCC 29212.

Compounds	Concentrations (*%*)	
	MIC	MBC
Catechin	0.625	1.25
Chlorhexidine	3.12	6.25
Fosfomycin	62.5	None

Chlorhexidine (CHx) & Fosfomycin have used a positive control.

**Table 4 molecules-26-06381-t004:** The binding affinity of catechin (**1**) and positive controls.

Ligand	Binding Affinity of Ligand-Protein Complex (kcal/mol)			
	MurA	GBAP	Gelatinase	Serine Protease
Catechin	−8.5	−5.2	−7.8	−7.0
Fosfomycin	−4.6	−3.1	−4.6	−3.8
Ambuic acid	−7.8	−4.5	−6.6	−6.5
Quercetin	−8.5	−5.2	−8.3	−6.9
Taxifolin	−9.1	−5.1	−7.9	−6.5

Quercetin, fosfomycin, ambuic acid, and taxifolin were used as positive controls.

**Table 5 molecules-26-06381-t005:** Hydrogen bond in catechin–MurA and positive controls.

Residues Binding at Ligand-Protein Complex				
Ligand	MurA	GBAP	Gelatinase	Serine Protease
Catechin	Ser162A, Gly164A, Asp305A, Ala297A	Arg15A, Ser33A, Val16A	His332A, Tyr343A, Glu329A, Asn298A	Asn145A, Asn262A, Asn268A, Leu212A
Fosfomycin	Arg120A, Asn23A, Arg371A, Asp305A	Arg28A, Gln31A, Lys26A	Arg384A, Glu329A, Asn298A, His419A	Asn214A, Asp147A
Ambuic acid	Lys22A, Arg91A, Arg120A	Ile36A	Arg384A, Glu352A	Asn99A, Ala146A, Ser143A, Asn214A, Asp147A
Quercetin	Ser12A, Gly164A, Glu188A, Phe328A	Glu45A, Gln43A, Trp40A	Asn298A, His332A, Glu336A, Tyr343A, Glu329A	Asn145A, Asn262A, Asn268A, Glu213A
Taxifolin	Ser162A, Gly164A, Asp305A, Asn23A, Arg397A, Asp49A, Lys22A	Asn35A, Arg15A, Ser33A, Gln31A	Asn351A, His332A, Asn298A, Glu329A, Trp301A	Arg46A, Ser83A, His195A

Quercetin, fosfomycin, ambuic acid, and taxifolin were used as positive controls.

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
