# Peer review of "Bio-Mechanism of Catechin as Pheromone Signal Inhibitor: Prediction of Antibacterial Agent Action Mode by In Vitro and In Silico Study"

_molecules, 2021, doi:10.3390/molecules26216381_

Round 1

Reviewer 1 Report

The authors of the manuscript demonstrated that catechin has potency as an antibacterial and anti-QS agent. It compound can be used as an alternative antibacterial agent with the most effective mechanism to prevent and control oral disease affected by antibiotic resistance. I have some comments:

  1. In the abstract and in the introduction, it is worth highlighting the scientific novelty of the manuscript.
  2. The analysis of Uncaria gambir Roxmb extract can be extended to other methods, such as ultra-performance liquid chromatography–quadruple–time of flight mass spectrometry (UPLC–QTOF–MS).
  3. Can, apart from catechin, other polyphenolic compounds contained in the extract also have effective antimicrobial activities?

Author Response

Dear, 

Thank you for your valuable review. The responds for your corrections and suggestion is in attached file.

Thank you

Reviewer 2 Report

I have found your article interesting and also in the field of my interest.

However, I would like to give you some suggestions.

  1. The author should show methods or scheme for bioactivity-guided isolation using a combination of chromatographic methods, together with HPLC or TLC data.
  2. The author should list raw NMR original spectra of "isolated" compounds.

Author Response

Dear, 

Thank you for your valuable review. 

The corrections and suggestion to your comments is in attached file.

Thank you

Reviewer 3 Report

The authors describe the isolation and study of catechin as a pheromone signal inhibitor.

The paper is well written, I would reconsider after addressing the the following comments:

1. In 2.2 the authors describe the assignment of compound 1 and give the list of peaks (lines 138-141). A small text for the 2D data (as reported in line 313) is needed. Moreover, Figure 5(1) doesn’t correspond to the proposed compound (see my comment 3).

2. Did they consider or verified the presence of only one isomer(e.g not (-) catechin and only (+)catechin or a mixtrure of them?). A comment would be useful.

3. In Figure 1, the 5, (1) is quercetin. 

4. In table 4, the authors report a bonding affinity of 8.1 for catechin-MurA. In their previous work “The potency of catechin from gambir (Uncaria gambir roxb.) as a natural inhibitor of mura (1uae) enzyme: In vitro and in silico studies” by Riyana, B., Huspa, D.H.P., Satari, M.H., Kurnia, D.,(2020) Letters in Drug Design and Discovery, 17 (12), pp. 1531-1537.

the authors provide a corresponding value of -8.5 Kcal/mol. A comment  for this difference would be useful.

Author Response

Dear,

Thank you for your valuable review.

The corrections to your suggestion is in attached file.

Thank you

Reviewer 4 Report

In my opinion, the manuscript can be accepted for publication in this journal after correcting few minor shortcomings:

- chemical composition of Uncaria gamibir extracts should be included, qualitative and quantitative data are missing in Result section, chemical analysis should be describe in Material and Methods section

- location where the plant material was collected should be specified in more detail

- unit for inhibition zone of chlorhexidine should be corrected (line 229)

- italic should be used for name of bacteria E. faecalis (lines 145, 152)

Author Response

Dear,

Thank you for your valuable review.

The revision according to your correction and suggestion is in attached file

Thank you

Round 2

Reviewer 1 Report

I accept the manuscript in its current version.